# Prospect theory, constant relative risk aversion, and the investment horizon

**Haim Levy**[⊙]**, Moshe Levy**[ID][⊙]*

The Hebrew University, Jerusalem, Israel

⊙ These authors contributed equally to this work.
* mslm@huji.ac.il

## Abstract

Prospect Theory (PT) and Constant-Relative-Risk-Aversion (CRRA) preferences have clear-cut and very different implications for the optimal asset allocation between a riskless asset and a risky stock as a function of the investment horizon. While CRRA implies that the optimal allocation is independent of the horizon, we show that PT implies a dramatic and discontinuous "jump" in the optimal allocation as the horizon increases. We experimentally test these predictions at the individual level. We find rather strong support for CRRA, but very little support for PT.

**Data Availability Statement:** The detailed experimental data is provided in the Supporting Information file (S5 Appendix).

**Funding:** The author(s) received no specific funding for this work.

## Introduction

The optimal asset allocation between stocks and bonds, and its dependence on the planned investment horizon, is one of the most central issues in financial economics. It is a question of great practical importance for investors saving for retirement. This paper theoretically analyzes and then experimentally investigates asset allocation choices corresponding to different investment horizons. We focus on the two main competing preference paradigms: expected utility with Constant Relative Risk Aversion (CRRA), which is widely employed in the economic literature, and Prospect Theory (PT), which has been a dominating paradigm in the last few decades. PT has been supported by numerous experimental studies (typically with relatively small stakes), and can explain phenomena which are hard to rationalize in the expected utility paradigm, such as the equity premium puzzle.

In this study we contrast PT with CRRA both theoretically and experimentally. We first theoretically analyze the changes in the optimal asset allocation as a function of the investment horizon for CRRA and for PT preferences, for general rate of return distributions. PT is analyzed with four alternatives suggested in the literature as the reference point. Based on this theoretical analysis we then experimentally study subjects' asset allocation choices as a function of the investment horizon in a controlled setting, where the two competing preference models have very clear and very different theoretical predictions. The experimental setup mimics a situation of large-stake decisions, such as savings for retirement.

It is well-known that CRRA preferences imply that the asset allocation should be independent of the investment horizon, when returns are i.i.d. *and portfolio revisions are allowed after each period* [1–3]. Quite surprisingly, in this study we show that even if revisions are not

**Competing interests:** The authors have declared that no competing interests exist.

allowed, which is the setup of our experiment, there is virtually no change in the optimal asset allocation as the horizon increases.

In contrast, PT preferences imply very different behavior. The PT value function weighs losses more heavily than gains, as captured by the property of loss aversion. Given a stock with a positive mean return, the probability of a loss typically decreases as the investment horizon increases. Thus, as the horizon increases, the stock becomes more attractive. This has led to the suggestion that myopic loss aversion, i.e. PT loss aversion combined with a short evaluation period, can explain the equity premium puzzle [4]. This prediction of PT, that the asset allocation to stocks should increase with the investment horizon is well-known, and has been experimentally tested by [5–11]. In this study we theoretically confirm that the PT optimal allocation to the stock indeed grows with the investment horizon, as suggested by previous studies. However, perhaps counterintuitively, we show that the optimal allocation to stocks does not grow gradually as the horizon increases. Instead, the optimal allocation "jumps" dramatically and discontinuously from 0% (or from about 30%, depending on the reference point) to 100% in the stock (or even more, if borrowing at the risk-free rate is possible). This property does not depend on the return distribution, the specifics of the PT value function parameters, or on the use of the cumulative prospect theory (CPT) decision weights. It is also robust to different modelling assumptions about the reference point, which can be either at the current wealth, the future value of wealth if invested in the risk-free asset, the future value of wealth if invested in the stock (namely, a stochastic reference point), or the expected value of wealth given the subject's choice, as suggested by [12]. We analytically derive and intuitively explain this peculiar behavior. With realistic PT and return parameters the jump to an optimal weight of 100% in stocks occurs at very short horizons of 2–3 years.

These very specific and very distinct predictions of the two competing models allow us to make clear-cut inference about preferences from the choices in the experiment. We find rather strong support for CRRA preferences, with 44% of the subjects being consistent with these preferences. In contrast, we find that only 6% of the subjects are consistent with PT preferences (for a very wide range of parameters, and alternative reference points).

Several previous studies have examined the implications of PT to asset allocation choice [13]. derive the optimal dynamic asset allocation for a PT investor in a continuous-time setting, where prices follow an Itô process and markets are complete. They find, as we do, that the optimal allocation to stocks increases with the investment horizon [14]. examines the trading volume implications of PT and finds a positive non-linear relation between trading volume and stock return volatility. He also derives the optimal asset allocation between a stock and a risk-free asset and finds that unless the stock's expected return exceeds a certain threshold (which depends on the investor's loss-aversion parameter) the investor's optimal proportion in the stock is zero, also consistent with our results. These papers investigate the asset allocation between stocks (or a stock index) and the risk-free asset. For a discussion of the optimal diversification *across stocks* for a PT investor see [15, 16]. The two papers most closely related to the present study are the experimental studies [5] (GP), and [6] (TTKS), who experimentally investigate the influence of the investment horizon on the *average* asset allocation, aggregated across all subjects. They find that when subjects are presented with longer horizons, on average they tend to increase their investment proportions in the stock, consistent with the predictions of myopic loss aversion. We find similar results regarding the average proportion. However, when we examine choices on the *individual* level, and compare them with the predictions of PT, we find very little support for PT.

The two main contributions of the present study relative to the existing literature are:

1) the theoretical identification of a discontinuous jump in the optimal asset allocation of PT investors as the investment horizon increases (or as the stock's return distribution gradually improves), and

2) the experimental investigation of this prediction at the individual level, rather than at the aggregate level.

Analysis of the results at the individual level yields not only a sharp test of the competing theories, but also allows an examination of the degree of heterogeneity among individuals. While the concept of a "representative agent" is widely employed, numerous studies have shown that heterogeneity may have fundamental economic consequences (see, for example, [17–22]).

In the next sections we analytically derive the optimal asset allocation choices implied by PT and by CRRA preferences (with no rebalancing).

## Optimal asset allocation for prospect theory investors

The Prospect Theory value function is given by:

$$V(x) = \begin{cases} -\lambda(-x)^{\beta} & \text{for } x \leq 0 \\ x^{\alpha} & \text{for } x \geq 0 \end{cases}, \tag{1}$$

where $x$ is the *change* of wealth relative to the reference point, the exponents $0 < \alpha \leq 1$ and $0 < \beta \leq 1$ imply risk-aversion for gains and risk-seeking for losses, and the constant $1 < \lambda$ is known as the loss-aversion parameter [23]. experimentally estimate $\alpha = \beta = 0.88$, $\lambda = 2.25$.

In what follows, we assume $\alpha = \beta$. There are two justifications for this assumption. First, experimental studies that estimate these parameters find that they are either exactly equal [23–25], or that they are very close to each other [26, 27]. Second, $\alpha \neq \beta$ theoretically implies accepting some fair symmetric gambles [28], in contradiction to the fundamental notion of loss aversion [29–31].

Regarding the reference point relative to which gains or losses are measured, there are four main variations suggested in the literature:

1) the future value of the current wealth if invested at the risk-free rate,

2) the current wealth,

3) the expected value of future wealth, given the investor's choice, and

4) the future (stochastic) value of the current wealth if invested in the stock.

All of these alternatives imply a dramatic jump in the optimal allocation to the stock as the horizon increases. Below we analyze alternative 1) in detail. The corresponding analysis for alternatives 2)-4), which are mathematically more complicated but yield similar results, are relegated to S1 Appendix.

## Case 1: Reference point at the future value of wealth invested at the risk-free rate

If the investment proportion in the stock is denoted by $w$, then the terminal wealth is:

$$\widetilde{W}_T = W_0[w\widetilde{R} + (1-w)R_f], \tag{2}$$

where $W_0$ is the initial wealth, $\widetilde{R}$ is the stochastic return on the stock (1+rate of return), and $R_f$

is the return on the risk-free asset (1+riskfree interest rate). When the reference point is the future value of the current wealth if invested at the risk-free rate, $W_0 R_f$, the change in wealth relative to the reference point is thus:

$$\tilde{x} = \tilde{W}_T - W_0 R_f = W_0 \left[ w\tilde{R} + (1-w)R_f - R_f \right] = W_0 w \left[ \tilde{R} - R_f \right]. \tag{3}$$

Denoting the return in *excess* of the risk-free rate by $r$ ($r \equiv R - R_f$), Eq 1 with $\alpha = \beta$ implies that the PT expected value is given by:

$$EV(w) = W_0^\alpha w^\alpha \left[ -\lambda \int_{-\infty}^{0} f(r)(-r)^\alpha dr + \int_{0}^{\infty} f(r) r^\alpha dr \right], \tag{4}$$

where $f(r)$ is the probability density function of excess returns. The optimum allocation to stocks is the value $w^*$ which maximizes Eq 4. This implies two corner solutions for the optimal asset allocation, depending on the sign of the square brackets in Eq 4: if the sign is positive the optimal investment weight in the risky asset is $w^* = 1$, and if the sign is negative we have $w^* = 0$ (it is assumed that no borrowing or short-selling are allowed, i.e that $1 \geq w \geq 0$). Namely, let

us define $A \equiv \dfrac{\displaystyle\int_{0}^{\infty} f(r) r^\alpha dr}{\displaystyle\int_{-\infty}^{0} f(r)(-r)^\alpha dr}$. Then, the optimal asset allocation is:

$$w^* = \begin{cases} 0 & if\ \lambda > A \\ 1 & if\ \lambda < A \end{cases}. \tag{5}$$

The above equation implies that the optimal asset allocation for a PT investor is a corner solution: either zero or one (in the special case where $A = 0$ the expected value is independent of the asset allocation, implying that the investor is indifferent to the allocation). Note that the optimal asset allocation is independent of the investor's initial wealth, $W_0$.

The optimal asset allocation given by Eq 5 depends on the preference parameters, $\lambda$ and $\alpha$, as well as on the excess return distribution $f(r)$. The excess return distribution and $\alpha$ determine the value of $A$. Thus, for a given return distribution and $\alpha$ there is a threshold value of the loss aversion parameter $\lambda$, below which the allocation to the stock is 100%, and above which the allocation is 0%. Intuitively, the lower the loss aversion, the more attractive the stock seems. If the distribution of excess returns, $f(r)$, is "improved" in the sense that the probability of a loss decreases and the probability of a gain increases, the value of $A$ typically increases (its denominator is decreased and its numerator is increased). This implies that the threshold $\lambda$ below which the optimal allocation to the stock is 100% increases, implying that the stock becomes the preferred choice for a wider spectrum of PT investors. As will become evident in what follows, this is exactly what happens when the investment horizon increases. The effect of the power exponent $\alpha$ on choice is more ambiguous, as it affects both the numerator and the denominator of $A$. The effect of $\alpha$ on the optimal allocation may thus depend on the exact distribution of $f(r)$, and in particular, on it's a-symmetry. Indeed, numerical calculation of the optimal asset allocation, reported in Table 2, reveals that the value of $\alpha$ has only a small effect on the optimal allocation.

Our focus in this study is on the effects of the investment horizon on the optimal asset allocation. As the horizon increases, the optimal allocation to the stock "jumps" from 0 to 1 (unless it is already 1 for the 1-period horizon). To see this, note that if the stock's

expected return is higher than the return on the risk-free asset, then, as the horizon increases, the probability that stock yields a higher return than the risk-free asset converges to 1 (indeed [32], empirically find that the probability of the stock return being higher than the bond return is about 66% for monthly returns, and it gradually increases up to 97%-99% for a five-year horizon). Thus, as the horizon increases $\int_{-\infty}^{0} f(r)dr$ approaches 0, and as $(-r)^{\alpha}$ is bounded (the rate of return on a stock can't be lower than -100%), the term $\int_{-\infty}^{0} f(r)(-r)^{\alpha}dr$ also converges to 0. Hence, when the horizon increases, the term $A$ increases to infinity, and the optimal allocation to the stock becomes 1 for any finite value of $\lambda$. This conforms to the profound intuition of [4, 33]: when the investment horizon increases the probability of a loss becomes smaller, and stocks become more attractive relative to the risk-free asset. The contribution of the present analysis is to show that this shift to stocks is not gradual: it occurs as a dramatic and discontinuous jump from an allocation of 0% to the stock to an allocation of 100%. Moreover, as we shall see below, one does not need a very long horizon to obtain this "jump"–for typical parameters it occurs at an investment horizon of 2–3 years.

The above analysis employs the objective probabilities. The results are qualitatively similar if one employs the PT (or CPT) decision weights instead. In this case, one should replace the objective p.d.f $f(r)$ in Eqs 4 and 5 with the PT or CPT decision-weight p.d.f., $f^{*}(r)$. This does not affect the result of a jump from 0 to 1 in the optimal proportion in the stock as the horizon increases (S2 Appendix shows this for the return distributions employed in our experiment).

We would like to emphasize that while the focus of the present study is the relationship between the optimal asset allocation and the investment horizon, the dramatic jump in the optimal allocation for a PT investor occurs when any of the return or preference parameters are varied. Namely, Eq 5 implies a dramatic jump from 0 to 1 (or vice versa) when the distribution of excess returns, $f(r)$ changes. This can happen as a result of an increase in the investment horizon, which is our main focus, but can also be a result of a change in the stock return distribution due to changing economic conditions, or to a change in the risk-free interest rate. For example, Fig 1 shows the optimal asset allocation as a function of the risk-free rate, $r_f$ ($r_f = R_f - 1$), for a PT investor with the parameters $\alpha = \beta = 0.88$, $\lambda = 2.25$ in [23], and the empirical stock return distribution, estimated by the annual returns on the S&P500 index during 1997–2016. The optimal allocation is computed by Eq 5. For comparison, the figure also shows the optimal asset allocation for an investor with a logarithmic utility function, found numerically. The figure shows that while the optimal allocation to the stock decreases gradually with $r_f$ for the investor with log preference (as no borrowing or short-selling are allowed, the allocation is bounded between 0 and 1), the PT investor switches dramatically from an allocation of 1 to the stock to 0, at critical value of $r_f = 0.035$ (i.e. 3.5%).

Let us now employ the above general analysis to the specific distributions employed in our experiment. In the experiment subjects are asked to allocate their investment between a risky stock and a risk-free bond. There are three different tasks, where the return distributions are different in each task. Table 1 provides the three tasks (the detailed experimental procedure is discussed below). In Task 1, $\tilde{R}$ can take two values, 0.9 or 1.3, with equal probabilities, and the risk-free rate is $R_f = 1.05$. These values correspond to excess returns on the stock of $r = 0.9 - 1.05 = -0.15$ and $r = 1.3 - 1.05 = 0.25$, respectively. Thus, the expected value in Task 1 is given

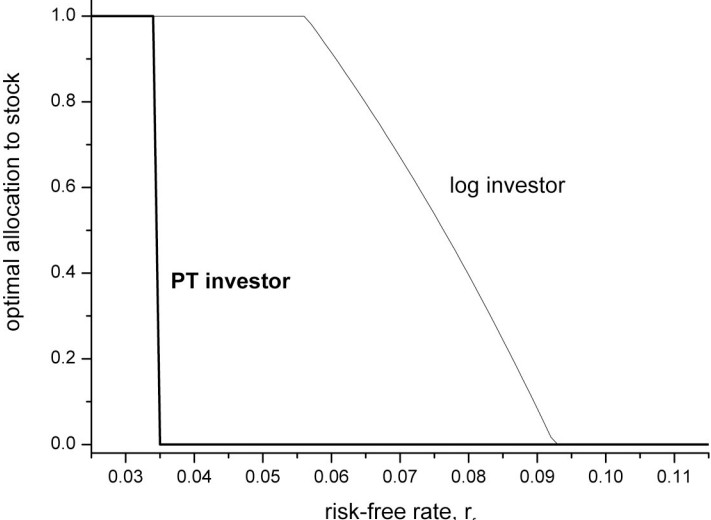

**Fig 1. The optimal asset allocation between the S&P500 index and a risk-free asset as a function of the risk-free interest rate $r_f$.** The empirical S&P500 annual returns during 1997–2016 are employed. The bold line shows the optimal allocation to the stock index for a PT investor with the parameters in [23]: $\alpha = \beta = 0.88$, $\lambda = 2.25$ given by Eq 5. The optimal allocation to the stock index drops dramatically from 100% to 0% once the risk-free rate reaches 0.035. In contrast, for an investor with a logarithmic utility function (thin line) the optimal allocation to the stock decreases gradually with the risk-free rate (note that the allocation is bounded between 0% and 100%, as no borrowing or short-selling are allowed).

by:

$$EV(w) = \frac{1}{2}(-\lambda)(W_0 w \cdot 0.15)^\alpha + \frac{1}{2}(W_0 w \cdot 0.25)^\alpha = \frac{1}{2}W_0^\alpha w^\alpha \left[ -\lambda 0.15^\alpha + 0.25^\alpha \right]. \quad (6)$$

**Table 1. The three tasks in the experiment are given below.** In each task the subject is asked to allocate his investment between the stock and the risk-free bond. Note that the return distributions and risk-free rates in Tasks 2 and 3 are the 2-period and 3-period distributions implied by the 1-period distribution and risk-free rate of Task 1, under the assumption of i.i.d. returns. The detailed experimental procedure is provided below.

**Task 1:**

| If you invest 100% in the stock | | | If you invest 100% in the risk-free bond | | |
|---|---|---|---|---|---|
| *Outcome* | *Rate of Return* | *Probability* | *Outcome* | *Rate of Return* | *Probability* |
| $90,000 | -10% | 50% | $105,000 | 5% | 100% |
| $130,000 | 30% | 50% | | | |

**Task 2:**

| If you invest 100% in the stock | | | If you invest 100% in the risk-free bond | | |
|---|---|---|---|---|---|
| *Outcome* | *Rate of Return* | *Probability* | *Outcome* | *Rate of Return* | *Probability* |
| $81,000 | -19% | 25% | $110,250 | 10.25% | 100% |
| $117,000 | 17% | 50% | | | |
| $169,000 | 69% | 25% | | | |

**Task 3:**

| If you invest 100% in the stock | | | If you invest 100% in the risk-free bond | | |
|---|---|---|---|---|---|
| *Outcome* | *Rate of Return* | *Probability* | *Outcome* | *Rate of Return* | *Probability* |
| $72,900 | -27.1% | 12.5% | $115,762 | 15.76% | 100% |
| $105,300 | 5.3% | 37.5% | | | |
| $152,100 | 52.1% | 37.5% | | | |
| $219,700 | 119.7% | 12.5% | | | |

This implies that there are two corner solutions for the optimal asset allocation, $w^*$:

$$w^* = 0 \text{ if } -\lambda 0.15^\alpha + 0.25^\alpha < 0, \text{ and :}$$

$$w^* = 1 \text{ if } -\lambda 0.15^\alpha + 0.25^\alpha < 0.$$

The threshold point that determines whether the optimal allocation in the stock will be 0 or 100% depends on the value function parameters $\alpha$ and $\lambda$. Namely, the optimal investment proportion in the stock, $w^*$, is given by:

$$w^* = \begin{cases} 0 & if \ \lambda > \dfrac{0.25^\alpha}{0.15^\alpha} \\ 1 & if \ \lambda < \dfrac{0.25^\alpha}{0.15^\alpha} \end{cases}. \tag{7}$$

Panel A of Table 2 shows the optimal investment proportion in the stock in Task 1 for different combinations of $\alpha$ and $\lambda$. For example, for the parameters in [23], $\alpha = 0.88$, $\lambda = 2.25$, we have $\lambda = 2.25 > \frac{0.25^{0.88}}{0.15^{0.88}} = 1.56$, and therefore the optimal proportion in the stock is 0 (see the shaded cell in Panel A of Table 2). In contrast, for the combination $\alpha = 0.88$ $\lambda = 1.50$, the optimal proportion is 1 (see the light-shaded cell in Panel A of Table 2).

While the experimental estimates of $\lambda$ are typically in the range 1.8–2.3 (see [34] for a recent meta-analysis) and $\alpha$ is typically estimated to be in the neighborhood of 0.9, Table 2 reports the optimal asset allocation for a much wider range of parameters. This allows us to examine the robustness of the results to the parameter values, and to address the issue of possible heterogeneity among PT investors (as the parameter estimates are only population averages).

The stock return distribution and risk-free rate in Task 2 are the 2-period return distribution and risk-free rate implied by those of Task 1, assuming that returns are i.i.d. In Task 2 the *EV* is given by:

$$EV(w) = W_0^\alpha w^\alpha \left[ -\frac{1}{4}\lambda \, 0.2925^\alpha + \frac{1}{2} \, 0.0675^\alpha + \frac{1}{4} \, 0.5875^\alpha \right], \tag{8}$$

and again we have two corner solutions. The optimal investment proportion in the stock in Task 2 is given by:

$$w^* = \begin{cases} 0 & if \ \lambda > \dfrac{\frac{1}{4} \, 0.5875^\alpha + \frac{1}{2} \, 0.0675^\alpha}{\frac{1}{4} \, 0.2925^\alpha} \\ 1 & if \lambda < \dfrac{\frac{1}{4} \, 0.5875^\alpha + \frac{1}{2} \, 0.0675^\alpha}{\frac{1}{4} \, 0.2925^\alpha} \end{cases}, \tag{9}$$

(which is a special case of Eq 5 with the return parameters of Task 2). Panel B of Table 2 provides the optimal investment proportion in the stock in Task 2 for various combinations of $\alpha$ and $\lambda$. If we take, for example, the [23] parameters $\alpha = 0.88$, $\lambda = 2.25$ we have:

$\lambda = 2.25 < \frac{\frac{1}{4} \, 0.5875^{0.88} + \frac{1}{2} \, 0.0675^{0.88}}{\frac{1}{4} \, 0.2925^{0.88}} = 2.40$, and hence the optimal investment proportion in the

stock is 1, i.e. 100%. Thus, a PT investor with the parameters in [23], $\alpha = 0.88$, $\lambda = 2.25$, is expected to switch from a proportion of 0% in the stock in Task 1, to 100% in the stock in Task 2. Panel C of Table 2 provides the optimal asset allocation in Task 3, determined by Eq 5 and the return distributions of Task 3 (which are the 3-period distributions implied by Task 1). Again, we see the same pattern of two corner solutions.

**Table 2. Optimal investment proportion in the stock for PT preferences with reference point at the future value of current wealth ($W_0 R_f$).**

| A: Task 1 | | | | | | | | | |
|---|---|---|---|---|---|---|---|---|---|
| Lambda\alpha | 1.00 | 0.98 | 0.96 | 0.94 | 0.92 | 0.90 | 0.88 | 0.86 | 0.84 |
| 1.00 | 1.00 | 1.00 | 1.00 | 1.00 | 1.00 | 1.00 | 1.00 | 1.00 | 1.00 |
| 1.25 | 1.00 | 1.00 | 1.00 | 1.00 | 1.00 | 1.00 | 1.00 | 1.00 | 1.00 |
| 1.50 | 1.00 | 1.00 | 1.00 | 1.00 | 1.00 | 1.00 | 1.00 | 1.00 | 1.00 |
| 1.75 | 0.00 | 0.00 | 0.00 | 0.00 | 0.00 | 0.00 | 0.00 | 0.00 | 0.00 |
| 2.00 | 0.00 | 0.00 | 0.00 | 0.00 | 0.00 | 0.00 | 0.00 | 0.00 | 0.00 |
| 2.25 | 0.00 | 0.00 | 0.00 | 0.00 | 0.00 | 0.00 | 0.00 | 0.00 | 0.00 |
| 2.50 | 0.00 | 0.00 | 0.00 | 0.00 | 0.00 | 0.00 | 0.00 | 0.00 | 0.00 |
| 2.75 | 0.00 | 0.00 | 0.00 | 0.00 | 0.00 | 0.00 | 0.00 | 0.00 | 0.00 |
| 3.00 | 0.00 | 0.00 | 0.00 | 0.00 | 0.00 | 0.00 | 0.00 | 0.00 | 0.00 |
| B: Task 2 | | | | | | | | | |
| Lambda\alpha | 1.00 | 0.98 | 0.96 | 0.94 | 0.92 | 0.90 | 0.88 | 0.86 | 0.84 |
| 1.00 | 1.00 | 1.00 | 1.00 | 1.00 | 1.00 | 1.00 | 1.00 | 1.00 | 1.00 |
| 1.25 | 1.00 | 1.00 | 1.00 | 1.00 | 1.00 | 1.00 | 1.00 | 1.00 | 1.00 |
| 1.50 | 1.00 | 1.00 | 1.00 | 1.00 | 1.00 | 1.00 | 1.00 | 1.00 | 1.00 |
| 1.75 | 1.00 | 1.00 | 1.00 | 1.00 | 1.00 | 1.00 | 1.00 | 1.00 | 1.00 |
| 2.00 | 1.00 | 1.00 | 1.00 | 1.00 | 1.00 | 1.00 | 1.00 | 1.00 | 1.00 |
| 2.25 | 1.00 | 1.00 | 1.00 | 1.00 | 1.00 | 1.00 | 1.00 | 1.00 | 1.00 |
| 2.50 | 0.00 | 0.00 | 0.00 | 0.00 | 0.00 | 0.00 | 0.00 | 0.00 | 0.00 |
| 2.75 | 0.00 | 0.00 | 0.00 | 0.00 | 0.00 | 0.00 | 0.00 | 0.00 | 0.00 |
| 3.00 | 0.00 | 0.00 | 0.00 | 0.00 | 0.00 | 0.00 | 0.00 | 0.00 | 0.00 |
| C: Task 3 | | | | | | | | | |
| Lambda\alpha | 1.00 | 0.98 | 0.96 | 0.94 | 0.92 | 0.90 | 0.88 | 0.86 | 0.84 |
| 1.00 | 1.00 | 1.00 | 1.00 | 1.00 | 1.00 | 1.00 | 1.00 | 1.00 | 1.00 |
| 1.25 | 1.00 | 1.00 | 1.00 | 1.00 | 1.00 | 1.00 | 1.00 | 1.00 | 1.00 |
| 1.50 | 1.00 | 1.00 | 1.00 | 1.00 | 1.00 | 1.00 | 1.00 | 1.00 | 1.00 |
| 1.75 | 1.00 | 1.00 | 1.00 | 1.00 | 1.00 | 1.00 | 1.00 | 1.00 | 1.00 |
| 2.00 | 1.00 | 1.00 | 1.00 | 1.00 | 1.00 | 1.00 | 1.00 | 1.00 | 1.00 |
| 2.25 | 1.00 | 1.00 | 1.00 | 1.00 | 1.00 | 1.00 | 1.00 | 1.00 | 1.00 |
| 2.50 | 1.00 | 1.00 | 1.00 | 1.00 | 1.00 | 1.00 | 1.00 | 1.00 | 0.00 |
| 2.75 | 1.00 | 1.00 | 1.00 | 0.00 | 0.00 | 0.00 | 0.00 | 0.00 | 0.00 |
| 3.00 | 0.00 | 0.00 | 0.00 | 0.00 | 0.00 | 0.00 | 0.00 | 0.00 | 0.00 |

The three shaded cells correspond to the parameters in [23] of $\alpha = \beta = 0.88$ $\lambda = 2.25$ (the light-shaded cell in Panel A corresponds to the case $\alpha = \beta = 0.88$ $\lambda = 1.50$).

A comparison of the three panels of Table 2 reveals that for most combinations of $\alpha$ and $\lambda$, a "jump" from 0% in the stock to 100% in the stock occurs as the horizon increases (as one moves either from Task 1 to Task 2, or form Task 2 to Task 3). Thus, with the annual return distribution employed in our experiment, for PT investors, increasing the horizon from 1 year to 3 years implies a jump from 0% in the stock to 100% in the stock (unless the optimal allocation to the stock is already 100% at the 1-year horizon).

Employing Cumulative Prospect Theory (CPT) decision weights instead of the objective probabilities yields similar results. The table in S2 Appendix shows the optimal asset allocation in Tasks 1–3 when the CPT decision weights are employed ($\gamma = 0.61$, $\delta = 0.69$, see [23, 35] for evidence supporting CPT decision weighting).

The alternative cases of a reference point at the current wealth, the expected future wealth given the investor's choice, and the future wealth if invested in the stock, yield a similar jump in the optimal asset allocation as the horizon increases. The jump is not necessarily from 0 to 1 in these cases, but rather, from some positive value (e.g. 0.3) to 1. These cases are analyzed in S1 Appendix.

## Optimal asset allocation for CRRA preferences

The Constant Relative Risk Aversion (CRRA) utility function is given by: $U(W) = \frac{W^{1-\alpha}}{1-\alpha}$, where $W$ denotes wealth, and $\alpha$ is the coefficient of relative risk aversion. Consider the asset allocation decision of a CRRA investor investing for a single period. The terminal wealth of the investor is given by: $\tilde{W} = W_0 \left[ w\tilde{R} + (1-w)R_f \right]$, where $W_0$ is the initial wealth, $\tilde{R}$ and $R_f$ denote the stock and risk-free bond returns, respectively, and $w$ is the proportion of wealth allocated to the stock. The investor's expected utility is thus:

$$EU(w) = \frac{W_0^{1-\alpha}}{1-\alpha} E\left[ \left(w\tilde{R} + (1-w)R_f\right)^{1-\alpha} \right]. \tag{10}$$

The optimal asset allocation for this investor, $w^*$, is the value that maximizes the expression $\tilde{W} = E\left[ \left(w\tilde{R} + (1-w)R_f\right)^{1-\alpha} \right]$ (or minimizes this expression, in the case of $\alpha > 1$).

Now, consider the same investor, who invests for two periods. The investor's terminal wealth is given by: $\tilde{W} = W_0(w_1\tilde{R}_1 + (1-w)R_f)(w_2\tilde{R}_2 + (1-w)R_f)$, where $\tilde{R}_1$ and $\tilde{R}_2$ denote the stock returns in periods 1 and 2, and $w_1$ and $w_2$ denote the allocation to the stock in the two periods. The investor's expected utility is given by:

$$EU(w) = \frac{W_0^{1-\alpha}}{1-\alpha} E\left[ \left(w_1\tilde{R}_1 + (1-w)R_f\right)^{1-\alpha} \left(w_2\tilde{R}_2 + (1-w)R_f\right)^{1-\alpha} \right]. \tag{11}$$

Because the stock returns are assumed to be i.i.d., this can be written as:

$$EU(w) = \frac{W_0^{1-\alpha}}{1-\alpha} E\left[ \left(w_1\tilde{R}_1 + (1-w_1)R_f\right)^{1-\alpha} \right] E\left[ \left(w_2\tilde{R}_2 + (1-w_2)R_f\right)^{1-\alpha} \right]. \tag{12}$$

(The period subscript of the stock returns can be omitted, because $\tilde{R}_1$ and $\tilde{R}_2$ are identically distributed). Both $w_1$ and $w_2$ are chosen so as to maximize the expression $E\left[ \left(w\tilde{R} + (1-w)R_f\right)^{1-\alpha} \right]$ (or minimize this expression, if $\alpha > 1$), hence we have $w_1 = w_2 = w^*$. The same analysis extends to any number of investment periods. Thus, when the asset allocation can be revised every period, the optimal asset allocation of a CRRA investor is independent of the investment horizon. This is a well-known result, that has also been extended to the case where the investor has intermediate consumption [1–3, 36].

In the setup employed in our experiment, which follows the setup in [5, 6], portfolio revisions are not possible. The logic for choosing this setup is explained in the next section. We show below that despite the fact that revisions are not possible, the optimal asset allocation is *almost* independent of the horizon for CRRA investors. Thus, quite surprisingly, not allowing for portfolio revisions does not change the optimal diversification strategy much, as almost the same optimal diversification is obtained with and without revisions.

The optimal investment proportion in the stock in each of the three tasks, for various values of the relative risk aversion coefficient $\alpha$, is given in Table 3. These optimal proportions are calculated numerically by solving for the proportion that maximizes expected utility. Namely, for each value of $\alpha$ and each task, we numerically find the optimal proportion $w^*$ that maximizes

**Table 3. The optimal allocation to the stock in the three tasks for CRRA investors.**

| Risk aversion parameter α | Optimal Investment Proportion in the Stock, $w^*$ | | |
|---|---|---|---|
| | Task 1 | Task 2 | Task 3 |
| 0.5 | 1.000 | 1.000 | 1.000 |
| 1.0 | 1.000 | 1.000 | 1.000 |
| 1.5 | 0.924 | 0.925 | 0.926 |
| 2.0 | 0.689 | 0.690 | 0.692 |
| 2.5 | 0.548 | 0.549 | 0.550 |
| 3.0 | 0.456 | 0.455 | 0.455 |
| 3.5 | 0.390 | 0.389 | 0.388 |
| 4.0 | 0.340 | 0.339 | 0.338 |
| 4.5 | 0.302 | 0.301 | 0.300 |
| 5.0 | 0.271 | 0.270 | 0.269 |
| 5.5 | 0.246 | 0.245 | 0.244 |
| 6.0 | 0.226 | 0.224 | 0.223 |
| 6.5 | 0.208 | 0.207 | 0.206 |
| 7.0 | 0.193 | 0.192 | 0.191 |
| 7.5 | 0.180 | 0.179 | 0.178 |
| 8.0 | 0.169 | 0.168 | 0.166 |

Note that even though the optimal proportion is not exactly identical across the three tasks, because there are no portfolio revisions, they are almost identical.

the expected utility $EU[\tilde{W}(w)]$, subject to the constraint $0 \leq w \leq 1$, i.e. with no borrowing or short-selling. Note that, as in the case of PT investors, the optimal asset allocation is independent of the initial wealth, $W_0$.

It is well known that for a given stock return distribution and for a given riskless interest rate, the investment proportion in the stock decreases with an increase in the degree of risk aversion. The result that is more relevant to our study, and that is not obvious, is that for a given value of $\alpha$, the optimal proportion is almost identical across all three tasks, i.e. it is almost invariant to the investment horizon, despite the fact that no revisions were allowed. For example, if we take the typical case of $\alpha = 2$, we see that the optimal investment proportion in the stock is 68.9% in Task 1, 69.0% in Task 2, and 69.2% in Task 3. This is a very slight increase in the optimal proportion, certainly below the resolution that can be detected experimentally (less than 3% of the subjects answered the questionnaire in resolution of fractions of %). Also, the slight influence of the horizon on the optimal proportion is not the same for all values of $\alpha$: for $\alpha \leq 2.5$ the proportion slightly increases with the horizon, while for $\alpha \geq 3$ it slightly decreases (see Table 3).

The explanation for the fact that the optimal proportion is almost constant, even when revisions are not possible, is as follows. First, note that the return distributions in Task 2 are just the distributions obtained by investing in Task 1 for two periods, assuming i.i.d. (and the returns in Task 3 are the 3-period returns). Suppose, for example, that an investor chooses a proportion of $w = 0.5$ in Task 2 (corresponding to the 2-period horizon), and suppose that the stock yields a rate of return of 69%, corresponding to a rate of return of 30% on the stock in both periods. In this case, the rate of return on the investor's portfolio is: $0.5 \cdot 69\% + 0.5 \cdot 10.25\% = 39.625\%$. In contrast, suppose that the investor invests $w = 0.5$ in the stock for 1 period, as in Task 1, and that after 1 period he rebalances his portfolio and again invests $w = 0.5$ in the stock for the second period. In this case, his rate of return in the first period is $0.5 \cdot 30\% + 0.5 \cdot .5\% = 17.5\%$, and his rate of return in the second period is again 17.5% (again,

the stock return is assumed to be 30% in both periods). Thus, after two periods the portfolio grows by 1.175 · 1.175 = 1.3806, i.e. the rate of return on the portfolio after two periods is 38.06%. This is a little lower than the return of 39.625% obtained with no rebalancing (this small difference is due to the fact that after the first period the investor who rebalances sells some of the stock to maintain $w = 0.5$, and thus misses some of the gain from the increase in the stock price in the second period). Table 4 provides a comparison the entire rate of return distributions with and without revisions for the case of $w = 0.5$. As the table reveals, the return distributions are not exactly identical, but they are very close. We should note that the case of $w = 0.5$ depicted in Table 4 is the one where the difference between the two distributions, with and without portfolio rebalancing, is the greatest. As the return distributions are very close, this implies that the optimal asset allocation is very similar whether portfolio revisions are possible or not.

In general, for CRRA preferences we have:

- When revisions are allowed, the optimal asset allocation of the investor is independent of the investment horizon.

- For *any* given asset allocation $w$, the distributions of terminal wealth are almost the same with and without revisions

$$\Downarrow$$

- The *optimal* asset allocation is almost the same with and without revisions

$$\Downarrow$$

- The optimal asset allocation is almost independent of the investment horizon even when revisions are not possible.

Indeed, Table 3 reveals that the optimal investment proportion in the stock with no revisions is almost constant across the three tasks. This result is quite general and does not hinge on the specific distributions given in our experiment: it holds for different return distributions, and for horizons much longer than the 3-period horizon employed in our experiment. For example, if one considers asset allocation between the S&P500 index and the 3-month T-bill, employing the empirical annual returns and assuming a 10-year investment horizon, the CRRA optimal asset allocation is very similar with and without revisions: for $\alpha = 2$ the optimal allocation to stocks is 91.1% when annual revisions are allowed, and 91.4% without revisions; for $\alpha = 4$ the optimal allocation to stocks is 47.6% when annual revisions are allowed, and 46.1% without revisions. The figure in S3 Appendix shows the optimal asset allocation with and without revisions, for alphas in the range 0.5–7.5, and provides more detail. As most empirical and experimental evidence suggest that the risk aversion parameter is around 1–2

**Table 4. The portfolio rate of return distribution for a 2-period horizon, with and without revision of the portfolio weights after the first period.**

| probability | Portfolio return without revisions | Portfolio return with revisions |
|---|---|---|
| | ($w = 0.5$) | ($w = 0.5$) |
| ¼ | -4.375% | -4.938% |
| ½ | 13.625% | 14.563% |
| ¼ | 39.625% | 38.063% |

The case shown is for $w = 0.5$, which is the case where the difference between the two distributions is maximal (see footnote 9).

(see [37] and references within), the figure reveals that in the relevant range of risk aversion the optimal weight in the risky asset with and without revisions is almost identical. Thus, the optimal asset allocation of a CRRA investor is almost the same whether he invests for 1 year or for 10 years, even if portfolio revisions are not allowed.

In the context of our experiment, this result yields a clear-cut prediction for the asset allocation behavior predicted for CRRA individuals: the investment proportion in the stock should be constant (or almost constant) across all three tasks.

## The experiment

The experiment was conducted in a classroom setting with pen and paper. It is composed of three main choice tasks. In each task there is a risk-free asset and a risky stock, with a given return distribution. In each task the subject is asked to allocate his investment between these two assets. The complete questionnaire is provided in S4 Appendix. The instructions that appear before each one of the three tasks are:

> *Suppose that you have decided to invest $100,000 either in a stock or in a risk-free bond or in any combination of these two assets. The possible outcomes at the end of the investment period are as given below. Please write the percentage of the $100,000 you choose to invest in each asset, where the sum of the two investment proportions should add up to 100%.*

> *Investment proportion in stock: _______ Investment proportion in risk-free bond: _______*

The three tasks are given in Table 1. Note that Task 2 and Task 3 are in fact the 2-period and 3-period versions of Task 1, under the assumption of i.i.d. returns. In other words, the stock's return distribution in Task 2 is the return distribution obtained by investing in the stock given in Task 1 for two periods. Similarly, the risk-free rate in Task 2 is the 2-period return of the risk-free asset of Task 1. In the same manner, Task 3 is the 3-period version of Task 1. These three tasks allow us to investigate asset allocation choices, and their dependence on the horizon. The tasks are not framed in terms of a 1-period, 2-period and 3-period choices, but rather as three different stand-alone tasks. Specifically, in Tasks 2 and 3 the subjects have no opportunity to revise their allocations after each period (as in [5, 6]). This is important, because as shown in [33], subjects are strongly affected by the information presented to them: if the actual investment horizon is 3 periods, but the information provided to the subjects is for the 1-period returns, subjects tend to treat the investment as a 1-period investment. Thus, to obtain preferences for a 3-period horizon, the 3-period return distributions should be presented, which is the setting of our experiment, as well as in the experiments in [5, 6]. This approach also avoids possible biases that investors may have in translating 1-period returns into longer-period returns [38]. In addition, the cash flows of the various choices are obtained immediately, hence there is no need to discount the returns and the subjects do not face anxiety corresponding to immediate and far-away risks. The experimental setup employed corresponds to the framework in which the theoretical predictions the optimal asset allocation of PT and CRRA investors are derived–the optimal allocation is based on the t-period return distributions, which are the distributions presented to the subjects.

The experiment is designed to resemble a situation of large-stake investments, such as investment of life-long saving for retirement. It is conducted with large hypothetical gains and losses (the maximum gain in Task 3 is $119,700 and the maximum loss is $27,100). This has obvious pros and cons. The advantage of the large-stake setting is that it allows us to employ large potential gains and losses, which resemble decisions about life-long savings involving payoffs in the magnitude of hundreds of thousands of dollars. In addition, this may make subjects perceive their decisions as more substantial, relative to the typical

experimental setup, where the magnitude of outcomes are only a few dollars. Most importantly, we wish to create a setup which resembles (under obvious limitations) the subjects' total portfolio asset allocation choice, rather than resembling a small-scale bet. As [39] argues, individuals may behave very differently when large stakes are involved, compared to situations with small or moderate stakes.

The drawback of the hypothetical large-stake setup is that one may argue that subjects are not sufficiently incentivized when payoffs are hypothetical. To overcome this difficulty, we add a control task that allows us to examine whether subjects "took the experiment seriously", and to screen-out subjects who either did not give careful consideration to the questions, or did not fully understand them. The control task is the last task of the experiment, Task 4 (see S4 Appendix). In this control task, the subject is asked to choose between two alternative investments, where one alternative (investment G) dominates the other (investment F) by First-order Stochastic Dominance (FSD). This means that any rational individual (any expected utility maximizer as well as any cumulative PT expected value maximizer) should choose the dominating investment G (for more information on First-order Stochastic Dominance, see [40]). In this task we find that 90% of the subjects chose the dominating investment. This provides a strong indication that the vast majority of subjects did give careful consideration to the experimental tasks, and that they understood the instructions. We report below only results for those subjects who answered the control task correctly, i.e. we use the control task to screen out subjects who either did not give careful attention to their answers or did not understand the instructions. However, as only 10% of the subjects gave a wrong answer to the control task, the results do not change much if we include all subjects in the analysis.

The experiment was conducted with a hard-copy questionnaire filled out by pen. Written instructions were given in the questionnaire, as well as provided verbally. There was no show-up fee. The original questionnaire composed of the four tasks discussed above is provided in S4 Appendix.

## Subjects

The subjects are professional investors, and economics/business students. We have four groups of subjects, from three different countries:

**Group 1.**  57 undergraduate students from China (Harbin Institute of Technology, HIT), average age 24.4, 34% male.

**Group 2.**  17 master students from Hong Kong (Hong Kong Baptist University), average age 28.2, 67% male.

**Group 3.**  61 master students (MBA and economics) from Israel (Hebrew University), average age 29.5, 75% male.

**Group 4.**  68 professional investors from Israel (Harel Mutual Fund Company), average age 37.2, 81% male.

There are a total of 203 subjects. Of these, 184 (90%) answered the control task (with FSD dominance, see S4 Appendix) correctly. Another two subjects failed to complete the entire questionnaire. Thus, we are left with 182 subjects who completed the questionnaire and answered the control task correctly. The choices across the groups of subjects from various cultures and with different experience in the capital market are quite similar, which increases the reliability of our results. Below we report the results combined across all four groups. S5 Appendix provides the results for each subject group separately, and the individual level data, including group affiliation.

## Experimental results

While our focus is the diversification choices at the individual level, the average investment proportions in the risky asset are important for two reasons:1) it allows us to compare our results with other studies' results, which report only the average investment weight in stocks 2) The average allocation to stocks is important, as these figures are relevant for asset pricing. We find the following average investment proportions in the stock in the three tasks (across all 182 subjects):

$$w_1 = 53.7\% \quad w_2 = 59.0\% \quad w_3 = 58.1\%,$$

where the sub-indices refer to the task number. The average proportion increases somewhat from Task 1 to Task 2, and slightly decreases from Task 2 to Task 3. While the increase in the average proportion from Task 1 to Task 2 is statistically significant (matched-pair t-value 2.44), the decrease from Task 2 to Task 3 is not (t-value -0.12). It is interesting to note that [6] observe exactly the same pattern for the average proportions: a significant increase in the average proportion allocated to the stock from the short horizon to the medium horizon, and a slight decrease from the medium horizon to the long horizon. The results are compared in more detail in the next section. In general, the average proportions do not change much across tasks. These average proportions *may* be consistent with most subjects having CRRA preferences, where the small variation in the proportions across tasks is due to a small group of subjects with preferences different than CRRA. However, these average proportions *may* also be consistent with many other possible models, including a scenario where some subjects have PT preferences. Thus, in order to reach more definitive conclusions about preferences one cannot rely only on the average proportions, and we therefore turn to analyze the results at the individual level, which is our main experimental focus.

The preceding analysis shows that CRRA preferences predict investment proportions that are practically constant across all tasks, i.e. $w_1 = w_2 = w_3$, regardless of the risk aversion parameter, namely for all CRRA individuals. In contrast, PT preference predicts a jump of the investment proportion in the stock to 100%, either in Task 2 or in Task 3, depending on the preference parameters. Table 5 summarizes the choice patterns predicted by the different models, and the number of subjects with investment allocation choices conforming to each pattern.

Pattern 1, where the investment proportions are exactly equal across all three tasks, is consistent with CRRA preferences. While there are slight differences in the theoretically optimal proportions across tasks (because there are no revisions, see Table 3), these differences are in the order of 0.1%. As almost none of the subjects provided their answers with this degree of detail (more than 97% of subjects answered in whole percentage numbers), we consider this pattern consistent with CRRA. As reported in the table, 59 subjects (32.4%) follow this choice pattern. For each of these subjects we calculated the relative risk aversion coefficient, $\alpha$, that corresponds to his/her choice. We find an average $\alpha$ value corresponding to these 59 subjects of 1.96, which is similar to values previously reported in the literature (see [37] and references within). The standard deviation of $\alpha$ is 2.54, indicating a large degree of heterogeneity, even within CRRA subjects.

25 of the 59 subjects following pattern 1 chose $w_1 = w_2 = w_3 = 1$, i.e. they chose to invest 100% in the stock in all three tasks. While this pattern is consistent with CRRA preferences with $\alpha \leq 1$ (see Table 3), it could, in principle, be also considered consistent with PT, with $\lambda \leq 1.5$ (see Table 2, Tables AI, and AII in S1 Appendix). However, we do not consider this choice pattern as supporting PT for two reasons. First, estimates reported in the literature of the loss aversion coefficient, $\lambda$, exceed 1.75, and typically also exceed 2 ([41, 42] estimate that $\lambda$ is in

**Table 5. Asset allocation patterns observed in the experiment.**

| Pattern | Stock investment proportion in the three tasks | Preference consistent with choice pattern | Number of subjects | (%) |
|---|---|---|---|---|
| 1 | $w_1 = w_2 = w_3 \neq 0$ | Exact CRRA | 59 | 32.4 |
| 2 | $w_{max} - w_{min} < 0.1$ | Approximate CRRA | 80 | 44.0 |
| 3 | $w_1 = w_2 = w_3 = 0$ | PT with reference point either at the future value of wealth $\lambda \geq 2.75$,* or at the expected value of wealth** | 4 | 2.2 |
| 4 | $w_1 = 0, w_2 = 0, w_3 = 1$ | PT with reference point at future value of wealth $3 \geq \lambda \geq 2.25$,* or at the expected value of wealth** | 0 | 0 |
| 5 | $w_1 = 0, w_2 = 1, w_3 = 1$ | PT with reference point at future value of wealth $2.25 \geq \lambda \geq 1.75$,* or at the expected value of wealth** | 2 | 1.1 |
| 6 | $0.35 > w_1 > 0.25$, $0.40 > w_2 > 0.30$ or $w_2 = 1$, $0.40 > w_3 > 0.30$ or $w_3 = 1$ | PT with reference point at current wealth $\lambda \geq 1.5$*** | 5 | 2.7 |

$w$ is the investment proportion in the stock, and the sub-index refers to the task number. Patterns 1 and 2 correspond to CRRA and approximate CRRA preferences, respectively. Patterns 3–6 correspond to PT and approximate PT preferences, for a wide range of value function parameters and for different assumptions regarding the reference point.

* see Table 2;

** see S1 and S2 Appendices;

*** see Tables AI and AII in S1 Appendix.

excess of 2. Tversky and Kahneman [23] estimate $\lambda$ at 2.25, and [43] estimate it as 1.8 [34]. conduct a meta-analysis and estimate the loss aversion coefficient to be between 1.8 and 2.1). While we should allow for heterogeneity in $\lambda$ across subjects, we would expect to have $\lambda > 1.5$ for most PT investors. Thus, according to PT most choices should be according to patterns 3–6 (see below), and only a small minority should be $w_1 = w_2 = w_3 = 1$. Observing only a few choices of patterns 3–6, implies that we are observing *only* the subjects at the left tail of the $\lambda$ distribution, but not the subjects at the bulk of the distribution with $\lambda > 1.5$. This does not seem reasonable. Second, while a small value of $\lambda$ is technically consistent with PT, this is in a somewhat degenerate sense, because $\lambda \approx 1$ actually implies CRRA preferences (or almost linear preferences if, in addition, $\alpha \approx 1$).

The second pattern (see pattern 2 in Table 5) allows for some small variation in the proportions across tasks, possibly due to "noise", or bounded rationality errors. We consider all patterns where the maximum difference between the three proportions is smaller than 10% (i.e. $w_{max} - w_{min} < 0.1$), as "approximately consistent" with CRRA preferences. We find that 80 subjects (44%) fall into this category. Note that the 59 subjects in pattern 1 are a subgroup of the 80 subjects in pattern 2, i.e. when we allow for small variations in the allocations, 21 subjects are added to the 59 subjects in pattern 1.

We turn now to the choice patterns predicted by PT. We do not confine ourselves to the specific PT parameters suggested by [23], but rather, we consider a wide range of possible parameters $\alpha$, and all possible values of $\lambda$ larger than 1.75.

For $\lambda > 1.75$ PT with a reference point at the future value of wealth predicts $w_1 = 0$ in Task 1, see Panel A of Table 2. The investment proportion in the stock then jumps to 1 at either Tasks 2 or 3, or remains at 0, depending on the exact values of $\lambda$ and $\alpha$ (see Panels B and C of Table 2). Pattern 3 (with weights 0, 0, 0), pattern 4 (with weights 0, 0, 1), and pattern 5 (with weights 0, 1, 1) of Table 5 correspond to these three cases. Pattern 3 is predicted for all PT investors with a reference point at the future value of wealth. In aggregate, only 6 subjects

(3.3%) follow one of these three patterns. Thus, even if we allow for a wide range of PT parameters, there is very little support for PT with a reference point at the future value of wealth.

When the reference point is at the current wealth, and $\lambda > 1.75$, $w_1$ is between 0.30 to 0.33 (see Panel A of Table A1 in S1 Appendix), $w_2$ is either 0.35 or 1, and $w_3$ is either 0.37 or 1. In order to allow for some noise, or bounded-rationality errors (as we allow for approximate CRRA in pattern 2), we allow for deviations from these values and define pattern 6 as: $0.35 > w_1 > 0.25$, $0.40 > w_2 > 0.30$ or $w_2 = 1$, and $0.40 > w_3 > 0.30$ or $w_3 = 1$. Notice that PT subjects with a reference point at current wealth should follow this pattern whether they employ the objective probabilities or the CPT decision weights (see the table in S2 Appendix). Only 5 subjects (2.7%) follow pattern 6. We should note that some cases of pattern 6 may also be consistent (and are a special case of) patterns 1 and 2. For example, a subject who chooses $w_1 = w_2 = w_3 = 0.35$ is approximately consistent with PT (because he is included in pattern 6), but is also perfectly consistent with CRRA. Thus, there is some overlap between the choice patterns, and such a subject would be included in patterns 1, 2, and 6. Of the 5 subjects following pattern 6, four are also consistent with approximate CRRA, i.e. they do not exhibit any "jump". Thus, even when we allow for a wide range of PT parameters we find very weak support for PT: in total, at most 6% of the subjects are consistent with PT. We say "at most" because some of the subjects classified as PT subjects also fall in the CRRA category.

The other subjects who do not fall into any of the categories in Table 5, display a variety of choice patterns. 38 subjects (21%) monotonically increase their investment proportion in the stock with the horizon, i.e. $w_1 < w_2 < w_3$ (but not as predicted by PT). For 20 subjects (11%) the investment proportion monotonically decreases: $w_1 > w_2 > w_3$. 29 subjects display non-monotonic proportions: for 18 subjects we have $w_1 < w_2 > w_3$, and for 11 subjects $w_1 > w_2 < w_3$. (There are a few subjects who do not fall into any of these categories, for example subjects with $w_1 = w_2 > w_3$ $w_1 > w_2 = w_3$, etc.). The complete results of individual-level choices are provided in S5 Appendix.

To summarize, as expected, individual's choices are heterogeneous. Yet, the most dominant choice pattern is pattern 1, where all proportions are exactly identical. 32.4% of the subjects are exactly consistent with CRRA, and 44% are "approximately consistent" with CRRA. *At most* 6% of the subjects are approximately consistent with PT.

### Relation to the existing literature

We find evidence supporting CRRA preferences for a relatively large proportion of the subjects. This finding is consistent with previous empirical and experimental findings [44]: concludes that the relative risk aversion is almost constant, implying CRRA [45]. find support for CRRA based on survey data regarding households' asset allocations [46–48]. find experimental evidence in support of CRRA [49]. find that the empirical consumption and return data support CRRA preferences.

We find very weak support for PT preferences: at most 6% of the subjects are consistent with PT, and about half of these 6% are also consistent with CRRA (4 of the 5 subjects following pattern 6). This may seem surprising, in light of the vast experimental support for PT. How can this result be reconciled with the extensive literature showing support for PT? We believe that the most plausible explanation is that individuals behave differently when faced with small or modest bets (as in most experimental studies supporting PT) and when faced with large stakes, as in the present experimental setup. The importance of the scale of the amount at stake on choices is best summarized by Rabin [39] who writes:

*Expected utility theory may well be a useful model of the taste for very large scale insurance. Despite its usefulness, however, there are reasons why it is important for economists to recognize how miscalibrated expected utility is as an explanation of modest-scale risk aversion.* (p. 1286)

Another possible explanation is that much of the experimental support for PT comes from experiments with either positive or negative prospects, rather than with mixed prospects. Indeed [50], find support for PT when employing prospects in the positive domain or in the negative domain separately, but find that most subjects contradict PT when mixed prospects are employed. Needless to say, most real-life decisions involve both potential gains and potential losses, and this is also the framework of the present study. It is also possible that PT is approximately consistent with aggregate results, while it does not provide a good description of choice at the individual level. This is what [51] find.

The two papers most closely related to the present study are [5] (GP), and [6] (TTKS). Both of these studies experimentally investigate the influence of the investment horizon on the *average* asset allocation. While there are some important differences between these two studies, they both find that when subjects are presented with longer horizons, on average they tend to increase their investment proportions in the stock. This is consistent with myopic loss aversion–when individuals with PT preferences observe short-horizon returns they are confronted with a large probability of a loss, and therefore they tend to avoid the stock; as the horizon increases, the probability of a loss decreases, and individuals tend to increase their investment proportions in the stock ([4, 33]). At the aggregate level, this is similar to what we observe here: the average proportion in the stock increases from 53.7% in Task 1 to 59% in Task 2 (and then slightly decreases at Task 3). However, as discussed above, these aggregate results may also be consistent with most subjects having CRRA preferences, and the relatively small increase in the average investment proportion being driven by a (potentially small) group of investors with non-CRRA preferences. TTKS report a large heterogeneity in subjects' behavior (see their Fig 1, p. 655). This implies that individual behavior may be very different than the averages reported. Thus, only by looking at decisions at the individual level we can reach more definitive conclusions regarding preferences. In the present study we examine not only whether the optimal investment allocation to the stock increases with the horizon, but also the exact way in which it changes. Thus, the two main distinctions between GP and TTKS and the current study is that we analyze choices at the *individual level*, and that we have very clear predictions about the *exact asset allocation values* implied by PT.

One key difference between GP and TTKS is that GP investigate asset allocation between a risky stock and a risk-free bond (as in the present study), while TTKS study the asset allocation between a risky stock and a risky bond (which is less volatile than the stock, but still involves risk). This has an important implication for the optimal asset allocation of a PT investor: when the bond is risk-free, a dramatic jump in the asset allocation is theoretically predicted. Indeed, in the GP setup PT also implies a jump from 0% to 100% in the stock. When the bond is risky (as in TTKS, and also in [4]) the optimal asset allocation may change more gradually. This is an advantage of the GP setup, and the setup employed here: it yields very clear and testable theoretical predictions.

It is interesting to note that TTKS find that: "*. . .subjects made exactly the same choices (on average) when they were allocating assets one period at a time or when they were committing themselves for 400 trials. . .*" (p. 656). It is well-known that this behavior is consistent with CRRA preferences when revisions are allowed. In the preceding analysis we show that even when revisions are not allowed, as in the final decision in the TTKS setup, and as in our experiment, CRRA implies almost the same behavior. Thus, this finding by TTKS is actually consistent with our results, namely, seems to lend support for CRRA.

## Conclusions

This study employs the setting of investments for different horizons to theoretically analyze the predictions of two main contending preferences: Expected utility with Constant Relative Risk Aversion (CRRA) preferences, and Prospect Theory (PT) preferences (with and without decision-weights, and with various alternative reference points). We then experimentally investigate whether subjects' choices conform to these predictions. The special feature of this setup is that it yields very clear predictions for the individual-level behavior of CRRA and PT preferences. We show that for CRRA preferences, regardless of the risk aversion parameter, the asset allocation should remain virtually constant at different horizons, even when revisions are not allowed. In contrast, we show that for PT preferences the optimal allocation to the risky stock should "jump" dramatically and discontinuously from 0% (or about 30%, depending on the reference point) to 100% (or even more, if borrowing is possible) as the horizon increases. It is important to note that this jump does not require a very long horizon, it already occurs at shifts from a one-year horizon to a two-year or a three-year horizon. These theoretical results are quite general: they do not depend on the specific return distributions, the specific PT parameters and the functional form of the value function, and whether objective probabilities or CPT decision-weights are employed. It is important to emphasize that the jump in the asset allocation to the risky asset is a fundamental characteristic of PT and loss-aversion, and it occurs even if the horizon does not change, e.g., if the stock distribution is fixed and the risk-free interest rate increases. Generally, we do not obtain such a jump in the expected utility framework.

Like in [5] and in [6], in our experiment each subject faced three choices, not knowing that they correspond to various horizons (or to a combination of several lotteries). Thus, by the selected choices we can investigate the horizon effect on asset allocation, neutralizing the effect of the timing of cash flows (see, for example, [52, 53]). We find that 32% of the subjects' choices are exactly consistent with the predictions of CRRA. If we allow for some bounded-rationality errors, this figure increases to 44%. We find that *at most* 6% of the subjects are consistent with PT, even if we allow for some bounded-rationality errors, a wide range of PT parameters, and different assumptions regarding the reference point. This is an upper bound, because about one half of the subjects consistent with PT are also consistent with CRRA.

This very weak support for PT is surprising, given the large number of experimental studies supporting PT. We suspect that this difference may be partly due to the fact that many of these studies investigate the domain of gains and the domain of losses separately, while we employ mixed prospects. More importantly, it is possible that loss aversion plays an important role in setups where gains and potential losses are modest (e.g. a few tens of dollars), but when larger amounts are at stake, such as lifelong savings, CRRA provides a much better model of preferences. In his famous paper [39], shows that CRRA (and expected utility in general) are not consistent with actual behavior "in the small" and "in the large": realistic risk aversion over modest outcomes implies absurd risk aversion over large outcomes. We complement this result by showing that PT is also not consistent with behavior both "in the small" and "in the large": while PT may explain choices over modest outcomes very well, in a setting of lifelong savings with large outcomes it yields very unrealistic predictions. It is perhaps time to develop unifying models of preference that merge these very different behaviors "in the small" and "in the large".

Our findings have several key implications. The first is about the equity premium puzzle [37]. Benartzi and Thaler [4] suggest that this puzzle may be solved by myopic loss aversion: PT preferences and a short horizon (or a high evaluation frequency). The lack of experimental support that we find for PT preferences "in the large" casts doubt on the

validity of this explanation. We must conclude, like [54], that the equity premium puzzle is yet to be solved.

The second implication of our results is about the important role of heterogeneity. We find that subjects are very different not only along one dimension (such as the degree of risk aversion), but rather they follow completely different patterns of the asset allocations as a function of the horizon (some monotonically increasing the allocation to the stock with the horizon, some monotonically decreasing it, and some changing it non-monotonically). As [17–22] and others have argued, one should be very careful about reaching conclusions about the aggregate behavior of very heterogeneous individuals.

Having said that, we find that CRRA is the preference class that best describes subjects' behavior when large outcomes are at stake. Thus, if one must choose a single preference to model choice "in the large", CRRA seems to be the best alternative, by far.

## Supporting information

**S1 Appendix. Extension to alternative reference points.**
(DOCX)

**S2 Appendix. Numerical results for PT investors with various reference points.**
(DOCX)

**S3 Appendix. Asset allocation for CRRA investors with and without portfolio revisions.**
(DOCX)

**S4 Appendix. The questionnaire.**
(DOCX)

**S5 Appendix. Detailed experimental results.**
(DOCX)

## Author Contributions

**Conceptualization:** Haim Levy, Moshe Levy.

**Data curation:** Haim Levy.

**Formal analysis:** Haim Levy, Moshe Levy.

**Investigation:** Moshe Levy.

**Writing – original draft:** Haim Levy.

**Writing – review & editing:** Haim Levy, Moshe Levy.

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
