## [Decision Letter · Decision Letter 0]

10 Feb 2021

PONE-D-21-00889

Prospect Theory, Constant Relative Risk Aversion, and the Investment Horizon

PLOS ONE

Dear Prof. Levy,

Thank you for submitting your manuscript to PLOS ONE. After careful consideration, we feel that the paper has merit but needs revisions. Therefore, we invite you to submit a revised version of the manuscript that addresses the points raised during the review process.

We look forward to receiving your revised manuscript.

Kind regards,

Roberto Savona

Academic Editor

PLOS ONE

Journal Requirements:

Reviewers' comments:

Reviewer's Responses to Questions

**Comments to the Author**

1. Is the manuscript technically sound, and do the data support the conclusions?

Reviewer #1: Partly

2. Has the statistical analysis been performed appropriately and rigorously? 

Reviewer #1: No

3. Have the authors made all data underlying the findings in their manuscript fully available?

Reviewer #1: No

4. Is the manuscript presented in an intelligible fashion and written in standard English?

Reviewer #1: Yes

5. Review Comments to the Author

Reviewer #1: The submitted paper can be divided in two sections. In the first part, the authors demonstrate 1) a CRRA implies an optimal allocation between a riskless asset and a risky stock independent of the time horizon; 2) a PT implies a “jump” in the optimal allocation increasing the horizon. In the second part, the authors search for experimental support on these two results finding that most of the agents is CRRA with only few subjects which behave consistently with PT (at most 6%).

I found the paper unbalanced in the explanation of the two theoretical approaches: authors deeply describe the theoretical implications of PT, whereas section 3 (CRRA) is not well developed. Since PlosOne is a transdisciplinary journal, this lack of explanation is a real problem for the understanding of the readers.

2.1. Some comments:

- Pg. 2; authors wrote that the jump “does not depend on the return distribution, the specifics of the PT value function parameters, or on the use of the cumulative prospect theory”. Can the authors provide some explanations on the parameters affecting the “jump” property?

- Pg. 6; If I have correctly understood, the excess of the risk-free rate is r ~, hence in equations (4)-(5) should be r ~ not r.

- Pg. 10; It is not clear to me what happen in case of equality in equation (7).

- Pg. 13; It is not clear to me how have been defined the range of the parameters. In these tables the authors allow for lambda equal to 1 but, in section 5, they state that lambda is greater than 1.75.

- Pg. 15, authors state that the optimal allocation using CRRA is independent of the initial wealth. If I am not wrong, this also applies for PT.

- Pg. 19, authors write that Task-2 and Task-3 are the 2-period and 3-period version of Task-1 (one period) even if they are presented as three different stand-alone tasks. They justify this statement quoting Thaler et al. (1997) but, if I am not wrong, in that manuscript the subjects know the different horizon time (monthly, yearly, 5-yearly).

- Related to this issue, it is not clear to me how the authors can investigate the change in the horizon presenting to the subject three stand-alone tasks without any reference on time. The three situations can be perceived as three completely different asset allocations due to the bounded rationality of the subjects.

- Pg. 20, the authors should provide a deeper explanation on the fact that 1) the large hypothetical gains and losses may be perceived as more realistic; 2) the subjects act consistently with these possible large gains/losses, i.e. they make choices carefully, if there no incentives (as some payments) in performing accordingly (the authors only introduce a control task).

- Pg. 21, the original questionnaire is not provided.

- Pg. 22, the individual level data is not provided.

- Pg. 28, Thaler et al. (1997) is not a one-shot decision, subjects made repeated decisions over different horizon.

- Pg. 32, “the second implication of our results is about the important role of heterogeneity. […] subjects are very different […] follow different patterns of the asset allocation”. There is a huge literature on agent-based and asset price, and their experimental application, which is completely absent in the introduction.

6. PLOS authors have the option to publish the peer review history of their article (what does this mean?). If published, this will include your full peer review and any attached files.

Reviewer #1: No

---

## [Author Response · Author response to Decision Letter 0]

18 Feb 2021

Please see the attached revision report.

---

## [Decision Letter · Decision Letter 1]

8 Mar 2021

Prospect Theory, Constant Relative Risk Aversion, and the Investment Horizon

PONE-D-21-00889R1

Dear Prof. Levy,

We’re pleased to inform you that your manuscript has been judged scientifically suitable for publication and will be formally accepted for publication once it meets all outstanding technical requirements.

Kind regards,

Roberto Savona

Academic Editor

PLOS ONE

Additional Editor Comments (optional):

Reviewers' comments:

Reviewer's Responses to Questions

**Comments to the Author**

1. If the authors have adequately addressed your comments raised in a previous round of review and you feel that this manuscript is now acceptable for publication, you may indicate that here to bypass the “Comments to the Author” section, enter your conflict of interest statement in the “Confidential to Editor” section, and submit your "Accept" recommendation.

Reviewer #1: All comments have been addressed

2. Is the manuscript technically sound, and do the data support the conclusions?

Reviewer #1: (No Response)

3. Has the statistical analysis been performed appropriately and rigorously? 

Reviewer #1: (No Response)

4. Have the authors made all data underlying the findings in their manuscript fully available?

Reviewer #1: (No Response)

5. Is the manuscript presented in an intelligible fashion and written in standard English?

Reviewer #1: (No Response)

6. Review Comments to the Author

Reviewer #1: (No Response)

7. PLOS authors have the option to publish the peer review history of their article (what does this mean?). If published, this will include your full peer review and any attached files.

Reviewer #1: No

---

## [Editor Report · Acceptance letter]

18 Mar 2021

PONE-D-21-00889R1 

Prospect Theory, Constant Relative Risk Aversion, and the Investment Horizon 

Dear Dr. Levy:

I'm pleased to inform you that your manuscript has been deemed suitable for publication in PLOS ONE. Congratulations! Your manuscript is now with our production department. 

Kind regards, 

on behalf of

Prof. Roberto Savona 

Academic Editor

PLOS ONE